# The Endocrine Disruptor Compound Bisphenol-A (BPA) Regulates the Intra-Tumoral Immune Microenvironment and Increases Lung Metastasis in an Experimental Model of Breast Cancer

**DOI:** 10.3390/ijms23052523

**Published:** 2022-02-25

**Authors:** Margarita Isabel Palacios-Arreola, Norma Angelica Moreno-Mendoza, Karen Elizabeth Nava-Castro, Mariana Segovia-Mendoza, Armando Perez-Torres, Claudia Angelica Garay-Canales, Jorge Morales-Montor

**Affiliations:** 1Laboratorio de Genotoxicología y Medicina Ambientales, Departamento de Ciencias Ambientales, Instituto de Ciencias de la Atmósfera y Cambio Climático, Universidad Nacional Autónoma de México, Mexico City 04510, Mexico; mi.palacios.arreola@gmail.com (M.I.P.-A.); karlen@atmosfera.unam.mx (K.E.N.-C.); 2Departamento de Biología Celular, Instituto de Investigaciones Biomédicas, Universidad Nacional Autónoma de México, Mexico City 04510, Mexico; angelica@correo.biomedicas.unam.mx; 3Departamento de Farmacología, Facultad de Medicina, Universidad Nacional Autónoma de México, Mexico City 04510, Mexico; mariana.segovia@facmed.unam.mx; 4Departamento de Biologia Celular y Tisular, Facultad de Medicina, Universidad Nacional Autónoma de México, Mexico City 04510, Mexico; armandop@unam.mx; 5Departamento de Inmunología, Instituto de Investigaciones Biomédicas, Universidad Nacional Autónoma de México, Mexico City 04510, Mexico; clausgaray@iibiomedicas.unam.mx

**Keywords:** breast cancer, metastasis, cytokines, tumor microenvironment, Bisphenol A, endocrine disruptors

## Abstract

**Simple Summary:**

The widely spread microplastic component and endocrine disruptor BPA is a hazardous material recognized for a long time. Here, for the first time, we demonstrated that BPA, administered into mice in a very specific developmental step of the animal (3 days post-natal), induces an increase in metastasis to the lung in the adult life, compared to the control or vehicle mice. In addition, of novelty, it is the analysis of the cytokine tumor microenvironment, which is the reason for the increased metastasis by BPA (BPA induce the increase in pro-metastatic cytokines).

**Abstract:**

Breast cancer (BC) metastasis represents the main physiopathology leading to poor prognosis and death. Bisphenol A (BPA) is a pollutant, classified as an endocrine-disrupting chemical compound with estrogenic properties, their exposure in the early stages of neonatal life leads to an increase in the size and weight of breast tumors and induces cellular changes in the tumoral immune microenvironment where cytokines play a key role. Thus, we used female BALB/c mice exposed neonatally to a single dose of BPA. Once mice reached sexual maturity, a mammary tumor was induced, injecting 4T1 cells in situ. After 25 days of injection, we evaluated endocrine alterations, cytokine expression, tissue alterations denoted by macro or micro-metastasis in the lung, and cell infiltration induced by metastasis. We found that BPA neonatal treatment did not show significant endocrine alterations. Noteworthy, BPA led to an augmented rate of metastasis to the lung associated with higher intratumoral expression of IL-1β, IL-6, IFN-γ, TNF-α, and VEGF. Our data suggest that cytokines are key players in the induction of BC metastasis and that BPA (an environmental pollutant) should be considered as a risk factor in the clinical history of patients as a possible inductor of BC metastasis.

## 1. Introduction

Breast cancer is the leading cause of death in productive women between 20–50 years old, and the most prevalent cancer in women worldwide. In 2020, 2.3 million women were newly diagnosed with this disease, and 685,000 deaths were registered globally [1]. Breast cancer is also an economic burden because it is the leading cause of lost disability-adjusted life-years (DALYs) worldwide among any other type of cancer. Breast cancer is a heterogeneous disease where tumors can localize in different areas in the breast like the lobules, ducts, and connective tissue [2]. Specifically, lobular carcinoma is classified as the most common and invasive subtype [3]. Early detection is crucial in achieving long-term survival. Unfortunately, breast cancer cells can migrate to distant sites along the body, especially the lung, liver, bone, and brain, in a process known as metastasis, which is the leading cause of death; less than 20% of breast cancer patients with distant metastasis survive after five years [4,5].

Tumor progression and metastasis are highly influenced by the tumor microenvironment (TME). On this site, communication among tumor cells, tumor stromal cells, and immune cells are essential components [6]. In the beginning, tumor cells and stromal cells secrete soluble factors such as cytokines, chemokines, and growth factors, modifying cell-cell or cell-ECM (extracellular matrix) interactions and disrupting the normal epithelial organization [7]. This intercellular communication requires a complex network between stromal and immune cells [8]. This organization favors the proliferation, migration, and differentiation of tumor cells, suppresses the immune cells, and degrade ECM, which sooner or later will lead to a more invasive tumor that can break the connective tissue and metastasize [9].

Breast cancer etiology is associated not only with levels of specific hormones or their receptors, but importantly with more general environmental factors. Human industrial activity has provoked a colossal release of environmental chemicals to the atmosphere for decades, many of them with an unknown toxic effect on human health. Additionally, several daily use products, like plastic food and beverage containers, sunscreen, cosmetics, and cleaning products, along many, contain toxic chemicals [10,11]. Moreover, several epidemiological studies have provided strong evidence that associates toxicants with an increased risk to develop cancer in later stages of life. The possibility to develop a more aggressive type of breast cancer coincides with landmark events. For example, changes during prenatal, pubertal, pregnancy, and menopausal periods, where breast tissue suffers several changes in structure and function, and is more susceptible to specific environmental chemicals [12].

The endocrine-disrupting chemicals (EDC) are very important environmental chemical compounds because they can affect the hormone balance and the endocrine system [13]. The mechanism of action of EDC relies on binding to hormone receptors such as estrogen (ER) or androgen (AR) receptors, where they interrupt the functions of endogenous steroid hormones. Bisphenol A (2,2-Bis propane), known as BPA, is a synthetic chemical widely used in daily used products, from polycarbonate plastics to epoxy resins and dental sealants, and it is contained in food packing, baby bottles, medical devices, and personal care products, among others [14]. BPA has been classified as an EDC with estrogenic character, since it can bind to estrogen receptors, triggering signaling pathways, even when its affinity is lower than the endogenous ligand, 17 β-estradiol [15].

BPA is a compound that can be easily released from the plastics due to incomplete polymerization or hydrolysis of the polymers that conform the material, its detachment can be induced by high temperatures, acidic conditions, or enzymatic processes [16]. The main source of exposure to BPA in humans and animals is through food and beverages contained in materials where detachment from the matrix has occurred and it can be ingested, inhaled, and introduced by dermal exposure, dental sealants, or injections [17]. Despite the Food and Drug Administration (FDA) and the European Food Safety Agency (EFSA), which calculated that the tolerable daily intake of BPA is 50 μg/kg/day, it has been estimated that exposure to BPA per food package was higher in children from 1–2 months of age [18]. Exposure to BPA at tolerable concentrations or below is related to unfavorable effects on the health of humans and rodents [11,19].

The nature and magnitude of BPA’s adverse effects depend on the dose, the course of exposure, and the developmental stage in which exposure occurs. Exposure of BPA can occur as early as during gestation, according to reports of BPA presence in amniotic fluid, fetal serum, and breast milk [20,21]. In this regard, there is an existing concern about the effects that BPA could exert on a developing organism, including the immune system [22,23,24]. Several studies indicate that estrogen and progesterone stimulate the expression of the vascular endothelial growth factor (VEGF) in breast cancer and tumors [25]. VEGF is a key angiogenic factor that stimulates endothelial cells to proliferate and migrate, allowing tumors to progress easily [26]. In breast cancer, VEGF expression is increased depending on the microenvironment compared with normal mammary glands [27,28].

Previously, we have shown that after 25 days of injection, mice exposed to BPA presented no major endocrine alterations, developed larger tumors, higher proportion of regulatory T lymphocytes, together with decreased expression of TNF-α, IFN-γ, and the M2 macrophage marker Fizz-1. Furthermore, the cytometric analysis revealed differences in the expression of estrogen receptor (ER-α) in T lymphocytes, macrophages, and NK cells, both associated with exposure to BPA and tumor development [29].

Therefore, we decided to assess whether exposure to BPA in a critical development period affects not only tumor size, but also lung metastasis and cytokine expression pattern in tumors. Our results demonstrated that BPA administered during the neonatal period evoked an increase in lung metastasis and intratumoral cytokine pro-inflammatory pattern during adult life.

## 2. Results

### 2.1. Endocrine Parameters

As previously shown by us [29], to assess the potential reproductive effects of a single 250 μg/kg bw BPA dose, puberty onset was determined by the age of vaginal opening. We observed that exposure of female mice to BPA did not alter puberty onset. Furthermore, BPA did not influence baseline serum levels of estradiol during the diestrus phase (data not shown).

### 2.2. Tumor Size and Weight

We observed a remarkable increase in tumor development in the BPA-exposed group. At 25 days after the inoculation of tumor cells, it was evident that the mice subjected to neonatal BPA exposure developed bigger tumors. In fact, we confirm and extend our previous findings that after measuring tumor weight, those found in mice exposed to BPA showed an 88% increase in weight compared to the unexposed and vehicle groups [29] (data not shown).

### 2.3. IL-1β, IL-4, IL-6, and IL-10 Intratumoral Expression Pattern

Figure 1 shows a representative set of immunofluorescence images corresponding to the intratumoral expression of IL-1β, IL4, IL-6, and IL-10 in the different experimental groups, namely: control, vehicle, and BPA. Table 1 shows the quantification of the four cytokines expressed by the mean fluorescence intensity of the previous images. The results demonstrated that the expression of IL-1β was higher in the BPA group than in the control and vehicle treatments (Figure 1 and Table 2). We also detected an IL-4 intratumoral expression pattern, shown in Figure 1, where it contained representative images of the immunofluorescence stains corresponding to the expression of this cytokine in the different experimental groups. When IL-4 was quantified (Table 2), the tumors belonging to the control or vehicle groups showed a higher expression of IL-4 than tumors from animals treated with BPA, where there is a clear decrease in its expression (Figure 1 and Table 2).

**Figure 1 ijms-23-02523-f001:**
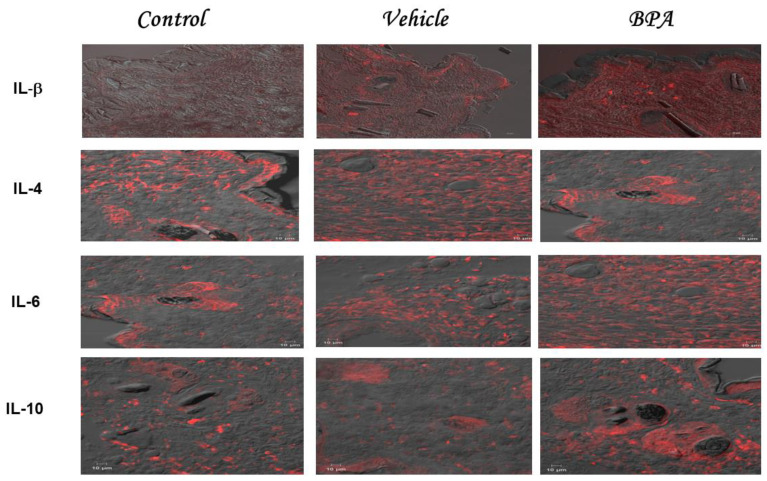
Intratumoral expression of cytokines. Representative images belonging to the intratumoral expression of IL-1β, IL-4, IL-6, and IL-10 corresponding to the three experimental groups, namely control, vehicle, and BPA, are shown. The staining was performed with rhodamine and pictures were taken in a confocal microscope, using Newarsky contrast, being red the cytokine expressed and gray the not stained tumor.

**Table 1 ijms-23-02523-t001:** Summary of the quantitation of immunofluorescence cytokine (IL1-β, IL-4, IL-6, and IL-10) staining in tumors. It shows the obtained MFI (Mean Fluorescence Intensity) (±SD) of the different experimental groups. It is highlighted the results that were considered statistically significant * *p* < 0.05, ** *p* < 0.01. Of note, only IL-4 decreased its expression in the BPA neonatally treated group.

Experimental Group	Immunohistochemical Staining of Cytokines (MFI)			
	IL1-β	IL-4	IL-6	IL-10
Control	183 ± 19.7	243 ± 10	256 ± 83	206 ± 83
Vehicle	206 ± 29	204 ± 67	208 ± 42	188 ± 42
BPA	531 ± 61 **	136 ± 64 *	541 ± 30 **	431 ± 30 **

Regarding the IL-6 intratumoral expression pattern, we observed that its expression was elevated in the control and vehicle groups as denoted by the increased fluorescence intensity; meanwhile in the BPA group, there was a significantly higher expression of IL-6 (Figure 1 and Table 1). It is noteworthy that fluorescence quantification produced a 2.9 increase in BPA-induced intratumoral levels of IL-6 compared to control or vehicle groups (Table 2). We also quantified immunoregulatory cytokine IL-10 in the tumors of all of the experimental groups. Intact and vehicle groups developed low fluorescence of IL-10. On the contrary, we found increased levels of fluorescence in the tumors of the BPA-treated mice (Figure 1). Regarding the quantification of IL-10 (Table 2), when we compared the expression of a control or vehicle, with BPA, we found a two-fold increase in the expression of IL-10. Notably, the different negative controls of each cytokine evaluated are compiled in Appendix A.

### 2.4. TNF-α IFN-γ and VEGF Intratumoral Expression Pattern

Intact and vehicle groups developed a mild fluorescence of TNF-α, with increased levels of fluorescence in the tumors of the BPA-treated mice (Figure 2). As for the quantification of TNF-α (Table 2) comparison between the expression of control or vehicle with BPA, it showed a 3-fold increase in TNF-α expression. Regarding intratumoral IFN-γ expression (Figure 2), the BPA-treated group showed higher levels of fluorescence compared to the control or vehicle groups, which showed mild levels of fluorescence (Figure 2). Quantification of fluorescence levels (Table 2), showed a significant 1.7-fold increase in the IFN-γ intratumoral expression in the BPA group, compared to the control or vehicle groups. As for VEGF, a pro-metastatic factor, we also evaluated its expression (Figure 2). Interestingly, tumors exposed to BPA presented higher levels of fluorescence than the control or vehicle groups (Figure 2). It is noteworthy that the presence of this cytokine was distributed along with the evaluated tumor court. Visual evaluation of VEGF matches its quantification (Table 2), since BPA-treated animals developed a 3-fold increase in VEGF expression compared to the control and vehicle groups. Notably, the different negative controls of each cytokine evaluated are compiled in Appendix A, which do not show any background or expression of cytokines.

**Figure 2 ijms-23-02523-f002:**
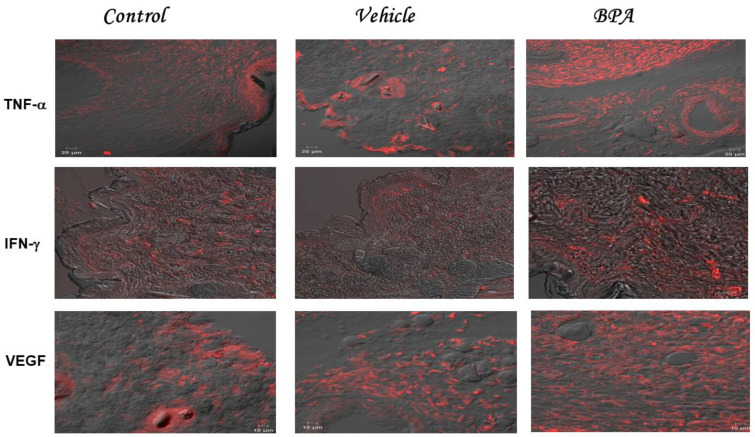
Intratumoral expression of cytokines. Representative images belonging to the intratumoral expression of TNF-α, IFN-γ, and VEGF corresponding to the three experimental groups, namely control, vehicle, and BPA, are shown. The staining was performed with rhodamine and pictures were taken in a confocal microscope, using Newarsky contrast, being red the cytokine expressed and gray the non-stained tumor.

**Table 2 ijms-23-02523-t002:** Summary of the quantitation of immunofluorescence cytokine (TNF-α and IFN-γ) and VEGF staining in tumors. It shows the obtained MFI (Mean Fluorescence Intensity) (±SD) of the different experimental groups. It is highlighted the results that were considered statistically significant * *p* < 0.05, ** *p* < 0.01, *** *p* < 0.001.

Experimental Group	Immunohistochemical Staining of Cytokines (MFI)		
	TNF-α	IFN-γ	VEGF
Control	186 ± 9.7	235 ± 10	206 ± 83
Vehicle	208 ± 59	204 + 67	198 ± 42
BPA	573 ± 61 **	366 ± 64 *	631 ± 30 ***

### 2.5. Macro Metastasis at the Pulmonary Level

In addition to assessing the effect of BPA on tumor growth and due to the migration of cancer cells in mammary tumors that have tropism towards the lungs, we decided to use this organ to evaluate the phenomenon of metastasis. Figure 3 shows a comparison of the lungs of all of the experimental groups. We observed noticeable damage in the lungs of the BPA-treated animals compared to the control and vehicle-treated groups, which seemed to collapse. In the image, although not very clear, there are macroscopic lesions on the surface of the lung. It is important to remark that at this time of sacrifice, even when we observed extensive damage in the lungs, the animals did not show apparent signs of pain or suffering.

### 2.6. Histological Examination of Lungs in Normal Animals without Tumors

Then, we decided to explore the histological damage of the lungs due to exposure to BPA in the female mice without tumors. Figure 4 shows the comparison of the microarchitecture of the lungs among experimental groups of the control, vehicle, and BPA-treated animals at different magnifications (4×, 20×, 10×, and 40×). We found no signs of inflammation, nor necrotic foci, and no new vascularization in any of the representative images shown. Of note, BPA treatment induced a slight inflammatory infiltrate into the lungs, as was judged at the 100× magnification in the center of the image. Moreover, the animals had no signs of disease (fur appearance, motility, mobility, food and water intake, and other behaviors were similar among non-tumor groups).

### 2.7. Histological Examination of Lungs and Micro Metastases

Micrometastasis of the control animals is shown at the pulmonary level at the same magnifications as previously mentioned to analyze the morphology and cellular infiltrate. We observed several different types of micrometastasis, namely multiple, bilateral, sharply outlined; rapidly growing; and more pleomorphic and necrotic sites in both groups. The lung metastasis was generally multiple, well-circumscribed, and spread rapidly. The metastasis looked like multiple discrete nodules in the periphery of the lungs or as lymphangitic carcinomatosis (peribronchial and perivascular patterns via the lymphatics). They rarely appeared as intra-lymphatic microscopic foci that cause pulmonary hypertension (Figure 5).

Eight weeks old mice were orthotopically implanted with tumor cell line 4T1. We found an increase in the inflammatory infiltrate, mainly by neutrophils, in the alveolar wall and neutrophilia in intact lung/4T1, in vehicle/4T1, and notably in mice neonatally exposed to BPA (BPA/4T1). These histopathological changes were associated with decreased alveolar ventilation in the mice implanted with 4T1, which presented subpleural and parenchymal metastasis. Clearly, the BPA treated group had more micrometastasis and high immune cell infiltration compared to the control 4T1 group. Vehicle-treated animals injected with 4T1 have similar histology patterns and metastasis as the control 4T1 group.

### 2.8. Histological Examination of the BPA-Treated Animals with Tumors

We decided to show here the analyses of the micrometastasis of the BPA-treated animals. The results showed that the animals exposed to BPA had severe histological changes (Figure 6). For example, we observed an alveolar collapse; remarkably, the alveoli were significantly infiltrated with neutrophils at the arterial level (center) and in medium vessels (arrows). The above can correlate with the air spaces (right) observed. On the other hand, the parenchyma had less density of neutrophils, which present good alveolar ventilation. The bronchioles (star) show focal epithelial detachment. Micrometastasis and neutrophils in the lung parenchyma were higher in animals exposed to BPA than in the control or vehicle groups.

Photomicrographs showed higher magnifications of the metastasis (100×). In this series of photomicrographs, we observed several micrometastatic sections; we found increased subpleural micrometastasis, with a broad presence of neutrophils and macrophages (Figure 6). Parenchymal (A−F) and subpleural (G−H) micrometastasis in the lungs of mice neonatally exposed to BPA and orthotopically implanted at eight weeks of age with tumor cell line 4T1. The infiltration of neutrophils in the alveolar wall and even inside a micrometastasis was noted (F). In the subpleural metastasis, we observed a mixed infiltrate of neutrophils and some macrophages (Figure 6).

## 3. Discussion

In the present work, we showed for the first time that a single neonatal administration of BPA induces remarkable histological alterations at the pulmonary level, correlated with important changes in the intratumoral expression of cytokines. As has been widely reported, the metastatic process can be promoted by the systemic and intratumoral production of cytokines [30,31,32]. For this reason, in a female mammary tumor model, we studied whether BPA treatment could affect the intratumoral expression of IL-1β, IL-4, IL-6, IL-10, TNF-α, IFN-γ, and VEGF by confocal microscopy immunofluorescence.

The effects of BPA on immune system cells have been widely reported; however, they vary depending on the model performed [33]. In vivo, the impact reported may seem contradictory, oscillating upon the animal species used, the dose, the administration route, the gender, the age, and the animal’s development stage in which BPA is administered [34,35]. Furthermore, many reports do not employ immune challenges for different immune components. Moreover, there is little information on the effects of BPA on the immune response during cancer context [34]. Most of the BPA effects on the immune system, point out to BPA as a pro-inflammatory molecule [36,37,38]. Our results support and extend this notion since the neonatal treatment of BPA clearly increased the pro-inflammatory cytokine profile.

Of particular interest is the finding that BPA decreases intratumoral expression of IL-4, a cytokine produced mainly by innate immune system cells and Th2 cells. IL-4 plays an important role in the humoral immune response against parasites and allergic antigens [39]. However, there are reports about the role of IL-4 in tumor growth, mediating increased proliferation and survival by promoting tumor-associated macrophage differentiation (TAM) towards an M2-like phenotype [40]. The decreased levels of IL-4 in BPA-treated animals, not necessarily is opposite to the role of IL-4 as a pro-metastatic molecule. This is not the only cytokine that promotes metastasis, also not all molecular pathways may be activated in every context.

In our present work, IL-4 appears to be not as relevant as the other cytokines found. Interestingly, we observed that the expression of IL-1β was higher in the BPA-treated group. IL-1β is a pleiotropic cytokine involved in inflammatory processes [41,42]. In different studies, elevated levels of this cytokine have been observed in breast cancer tumors and it has been proposed as a factor that promotes metastasis [43,44]. Thus, our present results confirm and extend the pro-metastatic role of this cytokine. Like IL-1β, IL-6 is also considered a master pro-inflammatory cytokine [45]. Interestingly, we found a higher intratumoral induction of IL-6 by neonatal exposure to BPA.

In breast cancer (BC), several studies have shown a positive relationship between the serum levels of IL-6 and the progression of the disease; the elevated concentration of this cytokine has been considered as a negative marker of prognosis in BC, independently of many factors, including hormonal status [46]. Particularly, IL-6 can promote metastasis by aberrantly activating the STAT3 pathway, supporting cancer stem cells (CSC) [47]. The activation of the IL-6/JAK/STAT3 pathway has also been implicated in the progression of BC [48]. Another pro-inflammatory molecule is TNF-α, a multifunctional pro-inflammatory cytokine that regulates different processes, such as inflammation, cell apoptosis, tumor growth, and cell invasion [49].

We also found an increased intratumoral expression of this cytokine after neonatal exposure to BPA. Regarding cancer, TNF-α promotes the invasion of breast tumor cells, as evidenced by in vitro experiments, up-regulating several genes associated with proliferation, invasion, and metastasis [50,51]. In addition, it has been shown that TNF-α can modulate the inflammatory role of macrophages, enhancing the production of VEGF [52]. Regarding BC metastasis, it has been well established that a critical regulator of this process is the VEGF family, which can be shaped by various receptors and ligands that confer a poor prognosis to patients with BC [53]. Specifically, VEGF-A has been closely related to neovascularization and angiogenesis in BC cells [54,55]. It is important to note that we did not analyze different types of VEGF in the tumor; however, we observed that the expression of VEGF-A was outstandingly modulated by neonatal exposure to BPA as it has a three-fold increase in response to it compared to the other experimental conditions. Our results on the implications of VEGF and metastasis are related to recent reports carried out in a BC lung metastasis mice model.

This work demonstrated the relevant relationship between VEGFR and metastasis-associated macrophages (MAMs), proving that this population is firmly implicated in the tropism of lung metastasis in this disease [56,57]. This idea agrees with our previous work, in which neonatal exposure to BPA modulated macrophage genes, which are involved in the polarization of the alternative phenotype with important implications in metastasis. In addition, the stimulating role of VEGF by different disrupting compounds, including BPA, has also been recently reported in BC cells [6,7].

Further to the above, a weakness of this work is that we did not evaluate the mechanisms through which BPA can modulate the expression of cytokines. Nevertheless, it has been reported that this endocrine disruptor, through its binding to the membrane ER, can impact the expression of various transcription factors, including PPARγ, which has pleiotropic actions at multiple levels and is importantly involved in the functioning of immune cells [58,59]. Different studies that investigate the mechanism through which BPA can modulate the expression of cytokines should be carried out to expand this information. Interestingly, the fact that we found an angiogenic/immunological modulation driven by BPA, which invites us to extend the cancer therapy options not only to breast cancer cells and cytokine signaling, but also to their endocrine disruptor counterpart.

## 4. Materials and Methods

### 4.1. Ethics Statement

Animal care and experimental practice were conducted at the Unidad de Modelos Biológicos (UMB) at the Instituto de Investigaciones Biomédicas (IIB), Universidad Nacional Autónoma de México. All experimental procedures performed in the animals were approved by the Institutional Care and Animal Use Committee (CICUAL, permit number 155); adhering to Mexican regulation (NOM-062-ZOO-1999), in accordance with the recommendations from the National Institute of Health (NIH) of the United States of America (Guide for the Care and Use of Laboratory Animals). Euthanasia of the experimental animals was performed humanely by overdose of inhaled Sevorane (Abbot, CDMX, México).

### 4.2. Animals

Mice of the syngeneic strain BALB/cAnN (H2-d) were purchased from Harlan México (Facultad de Química, UNAM, CDMX, México). The animals were housed at UMB with controlled temperature (22 °C) and 12-h light-dark cycles, water, and Purina LabDiet 5015 (Purina, St. Louis, MO, USA) chow ad libitum. After neonatal treatment, only female mice were used for experimentation.

### 4.3. Neonatal BPA Exposure

Although the main route of exposure to Bisphenol A (BPA, Sigma, St. Louis, MO, USA) is commonly oral, a subcutaneous injection was selected instead, since we did not observe differences between oral and subcutaneous routes in neonate mice. To resemble the human final gestational stage and aiming at the murine critical immune system development (T lymphocytes developmental window), the mice were exposed at postnatal day 3 (PND3). Briefly, 72 h after birth, female pups were identified by anogenital distance. Only female pups received treatment, although entire litters were assigned to experimental groups to avoid pup reallocation stress. The intact group received no neonatal treatment. The vehicle group received a dorsal subcutaneous injection of 20 µL of corn oil as the vehicle (Sigma, St. Louis, MO, USA). The BPA group received 250 µg/k body weight (bw) of BPA, dissolved in corn oil. Given that neonate rodents have minimal glucuronidation activity, which is the major metabolic mechanism for BPA clearance, this dose approximates to a brief, 5 day exposure to the FDA reference dose of 50 µg/k bw/day, but performed in a single administration, thus avoiding excessive manipulation stress. The pups were weaned at 21 days of age and placed in standard cages, with five mice per cage.

### 4.4. Assessment of Endocrine Parameters

*Vaginal opening.* From 25 days old onwards, the vaginal openings were examined by holding the mice in a dorsal restraint and using a light extension of the peri-vaginal skin. *Estrous cycle*. At 8 weeks old, the estrous cycles were assessed using a vaginal smear wash of 50 µL of saline solution (PiSA, Guadalajara, México), followed by Giemsa stain and light microscope observation. *Serum samples*. Corresponding to the diestrus phase, the serum samples were used to determine the estradiol levels, using the EIA DetectX^®^ Serum 17β-estradiol kit (Arbor Assays, Ann Arbor, MI, USA), according to the manufacturer’s protocol.

### 4.5. Cell Culture

The 4T1 cell line (ATCC^®^ CRL-2539) was kindly donated by Dr. Pedro Ostoa-Saloma and cultivated in RPMI 1640 medium (Sigma, St. Louis, MO, USA) supplemented with 10% FBS (ByProductos, Guadalajara, México), 2 mM glutamine, and penicillin/streptomycin (GIBCO, Grand Island, NY, USA). Subculture was performed at 70−80% confluency. After a second subculture, the cells were harvested and resuspended in 0.9% saline (250,000 cells/mL) for inoculation.

### 4.6. Mammary Tumor Induction

Upon sexual maturity (8 weeks old), mice from every exposure group were randomized into secondary experimental groups, i.e., control (without tumor induction) and 4T1 (tumor induction) groups. Mice assigned to 4T1 groups were treated as follows: Mice were anesthetized by the inhalation of a mixture of air and 5% Sevorane (Abbot, CDMX, México).

After low abdomen asepsis, the fourth nipple was located and 10^4^ 4T1 cells were introduced by a single injection into the mammary fat pad. Tumor growth was monitored for 25 days.

### 4.7. Histological Analysis of Lungs

The lung samples obtained from the experimental animals were fixed in 4% paraformaldehyde (J.T. Baker, CDMX, México), and were dehydrated and embedded in paraffin. Non-serial, longitudinal tissue blocks were cut into 4-m thick sections and mounted on poly-l-lysine coated slides (Sigma, St Louis, MO, USA). The histological analyses were performed with hematoxylin−eosin staining to identify neutrophils. The number of each cell type in the lung was calculated using a 40× objective. Several microscope fields, equivalent to 1 mm^2^, were analyzed for each mouse. The empty areas within the tissue were discarded using the software ImageJ (Version 1.39, Madison, WI, USA). A total area of 1 mm^2^ of lung was analyzed per group. The identification criteria were based on the morphological characteristics of the cells, which were also quantified according to each type.

### 4.8. Tumor Cytokine Immunofluorescence

Tumors from all groups were fixed in 4% paraformaldehyde for 48 h, washed with PBS, and stored in PBS containing 30% sucrose at 4 °C overnight. The next day, the samples were embedded in tissue freezing medium (Leica, Nussloch, Germany) and frozen at −70 °C (dry ice hexane bath). Serial sections of 20 μm thickness were obtained using a cryo-microtome (Leica, Buffalo Grove, IL, USA), placed on slides coated with poly-l-lysine (Sigma, St. Louis, MO, USA), and air-dried. The sections were treated with 1% Triton X-100 (Sigma, St. Louis, MO, USA), blocked with 1% albumin (BSA, Sigma, St. Louis, MO, USA), and incubated overnight at 4 °C with primary antibody diluted in 1:1000 BSA-PBS. Anti-mouse IL-1β, IL-4, and VEGF developed in rabbit, and anti-mouse IFN-γ, TNF-α, IL-10, and IL-6 developed in goat were used as primary antibodies (Santa Cruz Biotechnology, Dallas, TX, USA). Anti-rabbit IgG and anti-goat IgG conjugated with rhodamine (TRITC) were used as the secondary antibodies (ZYMED-Invitrogen Laboratories Inc., Grand Island, NY, USA). After rinsing in PBS, sections were incubated with a secondary antibody for 1 h at room temperature and were diluted in 1:200 BSA-PBS, washed in PBS, and embedded in anti-fading DAKO mounting medium (DAKO, Santa Clara, CA, USA). Tumor sections processed without the primary antibody were used as the negative controls.

### 4.9. Statistical Analysis

The general experimental design considers two independent variables: neonatal exposure (intact, vehicle, or BPA) and mammary tumor induction (control or 4T1). The data regarding tumor development and tumor microenvironment only consider the exposure variable, as all animals belong to 4T1 group. Data from two to three independent experiments were analyzed with Prism 6^®^ software (GraphPad Software Inc., San Diego, CA, USA) and charted as mean ± standard deviation. Data distribution normality was assessed via Shapiro−Wilk test. Thereafter, one-way ANOVA (*p* = 0.05) was performed, followed by a Tukey post-test. Differences were considered significant when *p* < 0.05, with the actual *p* value being stated in each figure legend. The data regarding cytokine expression consider both independent variables, and therefore a two-way ANOVA (α = 0.05) was performed, followed by a Holm−Sidak post-test, with the same significant difference criterion.

## 5. Conclusions

Metastasis is the major cancer pathophysiology and multiple biological interactions converge in common signaling pathways. The knowledge of this network leads to the possibility of developing specific therapeutic target drugs. Endocrine-disrupting compounds modulate endogenous hormone responses and cell functions. Although most studies have focused on their reproductive effects, their potential effects on immune cells, and even more, on the immune response towards cancer, should draw attention, given the expression of hormonal receptors by immune cells. Although several studies have evaluated the effects of BPA in the immune response, more studies are needed to elucidate the possible mechanisms through which these take place. Here, we exposed different molecular targets that should be blocked together, to offer a promising anti-metastatic drug regardless of the expression of hormonal receptors. We also want to highlight that not only cancer cells must be considered in the therapeutically strategy, but also the modification of the tumor microenvironment, and the metabolism of the surrounding cells play a key role in promoting or inhibiting the metastatic BC process, and this would result in a better patient clinical outcome. In addition, the immune cells and cytokines are key factors whose modulation would be important as adjuvant drug options in breast cancer metastasis.

## Figures and Tables

**Figure 3 ijms-23-02523-f003:**
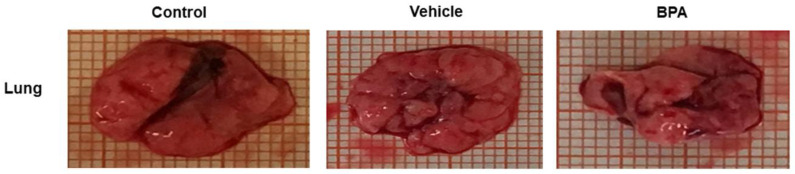
Macro metastasis at the pulmonary level. Representative images of the macro metastasis (tissue lesions) identified in the lungs belonging to the control, vehicle, and BPA-treated groups; millimetric grid as the background.

**Figure 4 ijms-23-02523-f004:**
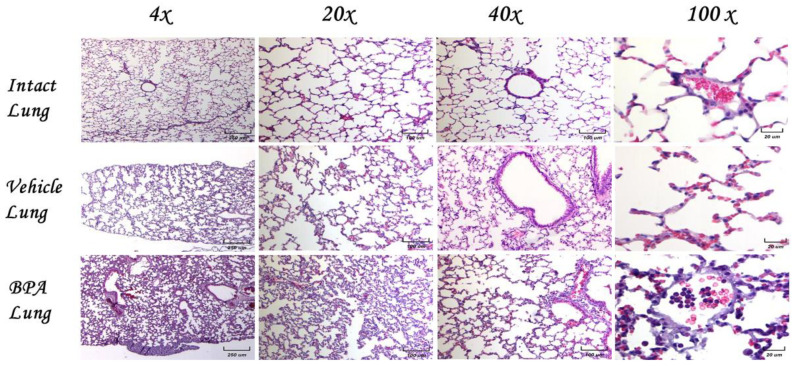
Histological examination of the lungs of female mice without tumors. Different magnification as indicated of lung tissues of different experimental groups. There is a normal histological appearance in intact or vehicle lungs, while BPA induces a slight infiltration of lymphocytes.

**Figure 5 ijms-23-02523-f005:**
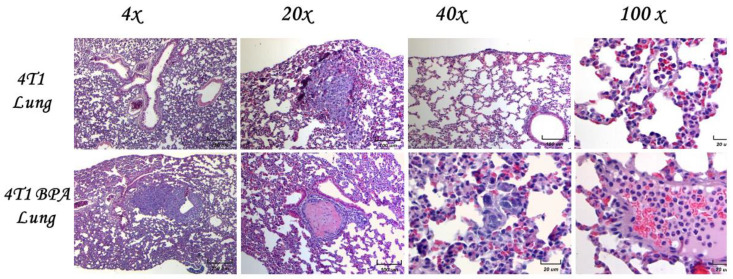
Histological examination of the lungs of female mice exposed to BPA without tumors. Representative images of the lungs in female mice exposed to BPA without tumor induction at different magnifications (4×, 20×, 40×, and 100×).

**Figure 6 ijms-23-02523-f006:**
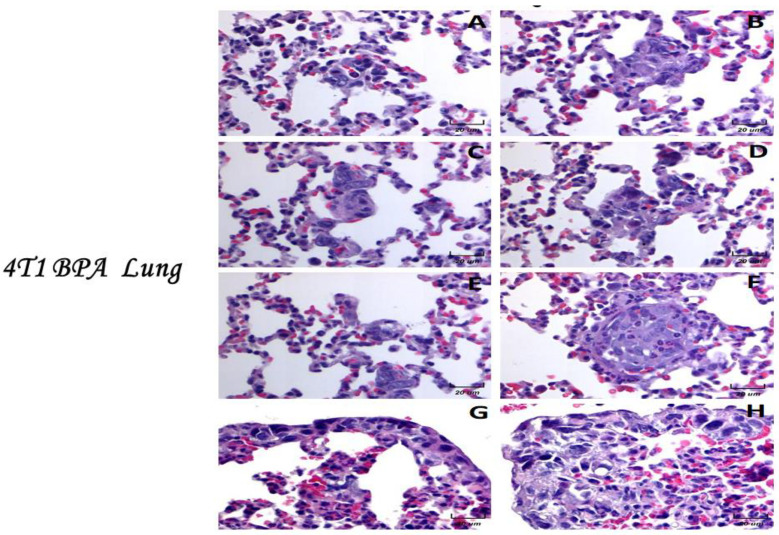
Histological examination of the BPA-treated animals with tumors. Parenchymal (**A**–**F**) and subpleural (**G**,**H**) micrometastasis in the lungs of mice neonatally exposed to BPA and orthotopically implanted at eight weeks old with tumor cell line 4T1. Representative images of the lungs of female mice with tumor induction neonatally exposed to BPA at 100× magnification.

## Data Availability

The datasets generated and analyzed during the current study are included in the present manuscript. They are also available from the corresponding author upon request.

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
