# Peer review of "The Endocrine Disruptor Compound Bisphenol-A (BPA) Regulates the Intra-Tumoral Immune Microenvironment and Increases Lung Metastasis in an Experimental Model of Breast Cancer"

_ijms, 2022, doi:10.3390/ijms23052523_

Round 1

Reviewer 1 Report

I am very glad to have a chance to read your work which gives concrete evidence of BPA for breast cancer. And this work seems quite good to prove the harness of BPA using in vivo system directly with full coloured H&E staining and immunohistochemical staining slides, which was actually impressive to me 

Your work seems to be very clear and give me powerful insight deal the BPA around you and me. Thank you.

Author Response

  1. I am very glad to have a chance to read your work which gives concrete evidence of BPA for breast cancer. And this work seems quite good to prove the harness of BPA using in vivo system directly with full coloured H&E staining and immunohistochemical staining slides, which was actually impressive to me. Your work seems to be very clear and give me powerful insight deal the BPA around you and me. Thank you.
  2. Thank you for your nice words and input. You did perfectly understood our message about the danger of BPA in a very specific developmental stage, and, what is the danger in particular during breast tumors. About the immunofluorescence technique, we implemented the Newarsky contrast, which gives a 3D appearence and contrast with red the positive expression, against the gray areas, which are negative. So, the effect of whatever molecule expression you are looking for, gets very clear. We appreciate your time and effort to review our manuscript. This comments make us feel that our work it is very well appreciatted into the international scientific community, particularly in the field of pollution. All english grammar was checked and corrected, so the minor spell erros were corrected.

We thank reviewer for the hard work put into revising our manuscript. As a result, a much better draft of our manuscript was generated. We hope that our responses will satisfy reviewers´concerns.

Reviewer 2 Report

Bisphenol A (BPA) is everywhere in our lives. It is present as a plasticizer on most plastics and packaging that we use everyday from bottled water, to other fruit juices, ..etc. BPA leaches quite readily from all these plastics and directly into our food and drinks. Further, FDA and other regulatory agencies put maximum intake at 50 ug/Kg per day. The authors emphasize that in reality, kids are exposed to much more than that.

BPA has great structural resemblance to tamoxifen, an estrogen receptor modulator. The authors claim that BPA may also act through HER receptor and trigger endocrine secretions.

They have injected female neonate mice at an early age with a dose of 250 ug/Kg BPA as a solution in corn oil, with a vehicle corn oil group control, as well as an untreated group. Mice were then challenged with cancer cells,  and then monitored release of several cytokines 25 days post injection. They have monitored endocrine alterations, in particular cytokine expression, and metastases to the lung. While no significant endocrine alterations occurred, BPA induced higher rate of metastasis to the lung  concomitant with higher intratumoral expression of IL-1ß, IL-6, IFN-gamma, TNF-alpha, and VEGF.

The effects showed unexpected results. The clear effect is that BPA plays quite a role in cancer development and metastasis in this model from a single injection at young age.

Perhaps an explanation for the choice of the dose is needed. It is not clear if the authors have run a pilot dose finding experiment which would have been very helpful for better experimental planning.

Importantly, as noted by the authors, the mice at such a young age do not have a developed metabolism that could eliminate BPA, and thus elimination which occurs in human subjects even at a young age, is absent here. The absence of metabolism which essentially equates to a high retention of PBA molecule and may translate to a rather high dose which may not relate to a daily intake of low content BPA which is essentially eliminated through glucuronidation. 

Author Response

  1. Bisphenol A (BPA) is everywhere in our lives. It is present as a plasticizer on most plastics and packaging that we use everyday from bottled water, to other fruit juices, etc. BPA leaches quite readily from all these plastics and directly into our food and drinks. Further, FDA and other regulatory agencies put maximum intake at 50 ug/Kg per day. The authors emphasize that in reality, kids are exposed to much more than that.
  2. Indeed that it is the case. Children are exposed to starting in the whole pregnancy, since BPA goes trought out the placenta, and the umbilical cord of the mother to them. Then, if breast feeded, trough out the mother´s milk. Eventually, they will be given bottles (mainly of plastic, and, BPA), and keep exposing, until they reach the puberty. Then, exposition continues. That it may ne one of the reasons why, chronic diseases as cancer, are going up, not down, even with the incredible advances on cancer reasearch.
  3. BPA has great structural resemblance to tamoxifen, an estrogen receptor modulator. The authors claim that BPA may also act through HER receptor and trigger endocrine secretions.
  4. Actually, we will slightly modify that sentence, since the EDC that has been demonstrated to act trough HER, it is BPS. However, BPA has been also claimed to possibly bind to HER. We are working with a group of people that has experience on structural biology to determine by docking, if BPA it is able to truly bind to HER. But evidence not so strong indicates that it may.
  5. They have injected female neonate mice at an early age with a dose of 250 ug/Kg BPA as a solution in corn oil, with a vehicle corn oil group control, as well as an untreated group. Mice were then challenged with cancer cells,  and then monitored release of several cytokines 25 days post injection. They have monitored endocrine alterations, in particular cytokine expression, and metastases to the lung. While no significant endocrine alterations occurred, BPA induced higher rate of metastasis to the lung  concomitant with higher intratumoral expression of IL-1ß, IL-6, IFN-gamma, TNF-alpha, and VEGF.
  6. Indeed, that was the case. We have new results (preliminary) that point out to an epigenetic effect of BPA on several cytokine promoters, reason why, with our scheme of application, no endocrine differences were seen, and, the effect was on those genes of cytokines and immune response. Particularly on pro-inflammatory and pro-metastasic cytokines.
  7. The effects showed unexpected results. The clear effect is that BPA plays quite a role in cancer development and metastasis in this model from a single injection at young age.
  8. At the begining we were also amazed of the effect of a single injection in the neonatal period in the mouse. Clearly, a single dose in the important developmental period of life, induces long lasting effects in the organism. Like I told you in upper lines, it is an epigenetic effect. I can tell you that, if we inject a single dose of BPA in the pre-puberal, puberal, young adult or old female mice, and, induce tumors there is not effect at all. Just in the neonatal period. Currently, we are working in administrating the BPA in the drinking water to the mother, and, then study the offspring. That will resemble better what happens in the human.
  9. Perhaps an explanation for the choice of the dose is needed. It is not clear if the authors have run a pilot dose finding experiment which would have been very helpful for better experimental planning.
  10. Thank you. Given that neonate rodents have minimal glucuronidation activity, which is the major metabolic mechanism for BPA clearance, this dose approximates to a brief, 5 days exposure to the FDA reference dose of 50 µg/k bw/day, but performed in a single administration, thus avoiding excessive manipulation stress to the pups. Although the main route of exposure to BPA is commonly oral, a subcutaneous injection was selected instead, since we did not observed differences between oral and subcutaneous routes in neonate mice. To resemble the human final gestational stage and aiming at the murine critical immune system development (T lymphocytes developmental window) mice were exposed at postnatal day 3 (PND3). All this information it is stated in Mat and Met.
  11. Importantly, as noted by the authors, the mice at such a young age do not have a developed metabolism that could eliminate BPA, and thus elimination which occurs in human subjects even at a young age, is absent here. The absence of metabolism which essentially equates to a high retention of PBA molecule and may translate to a rather high dose which may not relate to a daily intake of low content BPA which is essentially eliminated through glucuronidation. 
  12. It may be as you said. However, the 3-days age resembles to a 2-year human. Indeed it has been demostrated glucoronide of BPA in infants, however, the concentrations are much lower than the ones in puberal or adult individuals. However, your comment it is very well taken, and it may be considered in the future. I can tell you for instance, that we have co-related levels of parental compunds (not metabolites) of BPA, BPS and phtalathes in humans, and, levels are really high, and have nothing to do with the safe reported levels by FDA-NIH. If you are interested, check on this published paper: Mariana Segovia-Mendoza, Margarita Isabel Palacios-Arreola, Lenin Pavón, Enrique Becerril, Karen Elizabeth Nava-Castro, Omar Amador,and Jorge Morales-Montor. Environmental pollution to blame for depressive disorder? International Journal of Enviromental Research and Public Health. (2022). 19 (1) 1-13. DOI: /10.3390/ijerph19031737